# Leukotriene Signaling as a Target in α-Synucleinopathies

**DOI:** 10.3390/biom12030346

**Published:** 2022-02-23

**Authors:** Katharina Strempfl, Michael S. Unger, Stefanie Flunkert, Andrea Trost, Herbert A. Reitsamer, Birgit Hutter-Paier, Ludwig Aigner

**Affiliations:** 1Institute of Molecular Regenerative Medicine, Paracelsus Medical University, 5020 Salzburg, Austria; katharina.strempfl@pmu.ac.at (K.S.); michael.stefan.unger@everpharma.com (M.S.U.); 2Spinal Cord Injury and Tissue Regeneration Center Salzburg (SCI-TReCS), Paracelsus Medical University, 5020 Salzburg, Austria; 3QPS Austria GmbH, Neuropharmacology, 8074 Grambach, Austria; stefanie.flunkert@qps.com (S.F.); birgit.hutter-paier@qps.com (B.H.-P.); 4University Clinic of Ophthalmology and Optometry, Research Program for Ophthalmology and Glaucoma Research, Paracelsus Medical University, 5020 Salzburg, Austria; a.zurl@salk.at (A.Z.); h.reitsamer@salk.at (H.A.R.)

**Keywords:** leukotriene signaling pathway, Parkinson’s disease, dementia with Lewy bodies, α-synucleinopathy, Montelukast

## Abstract

Parkinson’s disease (PD) and dementia with Lewy bodies (DLB) are two common types of α-synucleinopathies and represent a high unmet medical need. Despite diverging clinical manifestations, both neurodegenerative diseases share several facets of their complex pathophysiology. Apart from α-synuclein aggregation, an impairment of mitochondrial functions, defective protein clearance systems and excessive inflammatory responses are consistently observed in the brains of PD as well as DLB patients. Leukotrienes are lipid mediators of inflammatory signaling traditionally known for their role in asthma. However, recent research advances highlight a possible contribution of leukotrienes, along with their rate-limiting synthesis enzyme 5-lipoxygenase, in the pathogenesis of central nervous system disorders. This review provides an overview of in vitro as well as in vivo studies, in summary suggesting that dysregulated leukotriene signaling is involved in the pathological processes underlying PD and DLB. In addition, we discuss how the leukotriene signaling pathway could serve as a future drug target for the therapy of PD and DLB.

## 1. Introduction

Parkinson’s disease (PD) and dementia with Lewy bodies (DLB) are two representatives of α-synucleinopathies, a group of age-related neurodegenerative conditions that are characterized by the excessive accumulation of α-synuclein (α-syn), mainly in neurons. With more than 6 million affected people worldwide, PD is the second most common neurodegenerative disorder after Alzheimer’s disease (AD) [1]. Widespread inclusions of α-syn in neuronal perikarya named Lewy bodies (LBs) and in neuronal processes called Lewy neurites (LNs), the consequent loss of dopaminergic neurons in the substantia nigra (SN) and a decrease in striatal dopamine are considered the key hallmarks of PD [2]. Clinically, PD manifests as progressive motor decline involving tremor, rigidity, and bradykinesia, as well as cognitive changes at later disease stages. The presence of LBs and LNs is also the defining pathological feature of DLB [3], a common type of degenerative dementia accounting for approximately 4–5% of the 50 million dementia cases worldwide [4,5]. In DLB, primarily the limbic and cortical brain regions are affected by neuronal damage, which is why DLB is usually more associated with cognitive impairments that precede parkinsonian symptoms compared to PD [3].

PD and DLB are multifactorial conditions with complex etiologies. Although the vast majority of cases occur sporadically and different environmental factors, such as exposure to herbicides/pesticides or heavy metals, strongly influence the susceptibility to developing α-synucleinopathies, several genetic risk variants have been identified. Some of these risk genes are shared between PD and DLB, as well as between familial and sporadic forms of the diseases [6,7,8,9,10] (Figure 1). Not only SNCA, which is the gene encoding for α-syn, but also GBA encoding β-glucocerebrosidase (GCase) is a mutual genetic risk factor of PD and DLB [6,11]. The enzyme GCase is active in lysosomes, where it catalyzes the cleavage of glucocerebrosides into glucose and ceramide. Reduced GCase activity is linked to the increased accumulation of its substrates and decreased lysosomal function, which may facilitate the aggregation of α-syn [6]. Mutations associated with PD, in addition to the α-syn gene, commonly occur in genes encoding for proteins that are part of signal transduction cascades and mitochondrial quality control pathways such as the leucine-rich repeat kinase 2 (LRRK2), Parkin, PTEN induced kinase 1 (PINK1) and DJ-1 [9,12]. An increased risk for developing DLB may be attributed to the apolipoprotein E (APOE) [6]. Under physiological conditions, APOE mediates the transport of cholesterol and other lipids to neurons, but its ε4 allele (APOE4) was found to exacerbate α-syn pathology [13].

As the pathomechanisms underlying PD and DLB are still not fully understood, therapeutic targets are lacking and curative therapies do currently not exist. Thus far, only symptomatic treatment options are available for PD and DLB. Most commonly, dopaminergic drugs such as the dopamine precursor levodopa or the dopamine agonist pramipexole are used [14]. Further approved symptomatic therapies against PD include monoamine oxidase B (MOA-B) and catechol-o-methyltransferase (COMT) inhibitors for suppressing the metabolism of endogenous dopamine, as well as deep brain stimulation (DBS) [15]. For the management of DLB, cholinesterase inhibitors, which improve cholinergic activity, may be used [16].

Mechanisms that have previously been proposed to contribute to α-syn aggregation and neurodegeneration in PD and DLB include oxidative stress, disruption of mitochondrial homeostasis, autophagy/lysosomal and synaptic dysfunction, endoplasmic reticulum (ER) stress and neuroinflammation [10,17] (Figure 2). More recently, a meta-analysis of human central nervous system (CNS) microarray transcriptomic studies identified the downregulation of neuronal energy metabolism, failure in protein degradation and the upregulation of predominantly microglial and astrocytic inflammatory responses as the central modulators of neurodegeneration [18].

In addition, the possible contribution of inflammatory signaling via leukotrienes, a family of potent lipid mediators, to the development of different neurodegenerative diseases has come to the fore (reviewed in [19,20,21]).

This review thus addresses molecular changes associated with the pathogenesis of PD and DLB and discusses the potential role of the leukotriene signaling pathway as an attractive target for the therapy of α-synucleinopathies and related proteinopathies.

## 2. The α-Synuclein Protein and Lewy Body Formation

Alpha-synuclein is a 14 kDa cytosolic protein that is abundantly expressed in neurons and, to a lesser degree, in glial cells throughout the CNS [22]. The protein consists of three distinct domains: an amphipathic N-terminal region, a central hydrophobic non-amyloid-β component region and an acidic C-terminal region. As the native structure of α-syn remains largely undefined, its exact function is not completely clear. Under physiological conditions, α-syn likely exists in a dynamic equilibrium between a stable homotetramer and an unfolded monomer [23]. It is postulated that interactions of α-syn with phospholipids promote the formation of a more stable N-terminal α-helical structure, suggesting a possible membrane-associated function [24]. The interaction of α-syn with membranes might influence membrane shape and induce curvature to facilitate synaptic vesicle budding and vesicle trafficking [25]. Apart from this, α-syn might regulate the release of dopamine and other neurotransmitters by controlling the formation of presynaptic SNARE (soluble N-ethylmaleimide-sensitive factor attachment protein receptor) complexes [26].

The conformational change of a monomer into a β-sheet structure causes the transition of the physiologically soluble α-syn into toxic insoluble deposits. In the β-beta sheet conformation, α-syn monomers stack together in a parallel fashion, forming first the protofilament and subsequently the mature fibril [27] (Figure 2).

Even though pathogenic variations of the SNCA gene are rare, several missense mutations, such as A30P, E46K, H50Q, G51D and A53T, are already identified in families throughout the world [6,9]. Nevertheless, multiplications of the entire coding region of SNCA are more frequently reported in familial but also sporadic cases of PD and DLB and are linked to propagating the assembly of toxic α-syn aggregates [6,9]. Furthermore, α-syn undergoes different post-translational modifications (PTMs). Certain PTMs might influence the accumulation process by affecting the membrane-binding ability of α-syn or by leading to aggregation-prone conformational changes [28]. In particular, a phosphorylation at serine residue S129 is the most commonly observed PTM in LBs [29].

Recent findings suggest that pathological neuronal α-syn inclusions may further drive the propagation of LB pathology through cell-to-cell transmission between interconnected brain regions [30].

## 3. Mitochondrial Dysfunction in Neurons

Neuronal cell death, the fundamental pathological feature of α-synucleinopathies, may directly be affected by the aggregation of α-syn and its interaction with mitochondrial structure and function. Paillusson et al. have shown that α-syn is a direct binding partner for the vesicle-associated membrane protein-associated protein B (VAPB) of the ER [31]. Together with protein tyrosine phosphatase-interacting protein 51 (PTPIP51), which is located in the outer mitochondrial membrane, VAPB couples mitochondria to the ER. This association enables the transfer of ER-stored Ca^2+^ to mitochondria via inositol 1,4,5-trisphosphate receptors (IP3Rs) of the ER membrane and the mitochondrial voltage-dependent anion channel (VDAC). Due to their high energy consumption, neurons are especially reliable on properly functioning Ca^2+^ exchange as ATP production is regulated by Ca^2+^ levels [32,33]. In the presence of overexpressed wild-type or mutant α-syn, the VAPB–PTPIP51 tie was shown to become disrupted, which loosens the ER–mitochondria contact and impairs the IP3R-mediated delivery of CA^2+^ to mitochondria [31]. The consequent reduction in mitochondrial ATP production in neurons seems to be a driver of neurodegenerative events.

Moreover, α-syn is able to reach the intermembrane space or even the matrix of mitochondria through translocation, most likely mediated by the TOM complex [34]. Previously, it has been shown that the binding of oligomeric α-syn to inner mitochondrial membranes impacts respiratory complex I activity [34,35] and induces oxidative events of the ATP synthase subunit β [36]. Together, these findings suggest that the localization of α-syn to mitochondria entails membrane depolarization events, leads to the excessive production of reactive oxygen species (ROS) and facilitates the opening of the permeability transition pore (PTP), resulting in mitochondrial swelling and increased cytochrome c release, which eventually triggers apoptosis in neurons [34,35,36].

Furthermore, α-syn oligomerization and the expression of peroxisome proliferator-activated receptor gamma coactivator 1α (PGC-1α), the master regulator of mitochondrial biogenesis, were found to influence each other negatively in PD [37]. Consequently, the downregulation of PCG-1α often observed in PD might contribute to α-syn-mediated toxicity and the detrimental effects on neuronal survival [37].

## 4. Dysregulated Protein Clearance in PD and DLB

The effective removal of misfolded and dysfunctional proteins, as well as of deleterious protein deposits, is crucial for the maintenance of neuronal cell homeostasis. Under physiological conditions, protein clearance is conducted by two main intracellular pathways: the ubiquitin–proteasome system (UPS) and the autophagy–lysosome pathway (ALP) [38]. Malfunctions in these pathways are linked to neurodegenerative processes in both PD and DLB, and increasing evidence suggests the involvement of α-syn in the impairment of these two protein degradation systems [38].

Previously, by applying in vitro proteasomal activity assays, researchers were able to show that aggregated α-syn inhibits proteasomal function, presumably through binding to the Rtp5/6S’ protein of the 19S regulatory particle (19S RP) [39]. The results of enzymatic assays in postmortem brain tissue are in line with these findings: decreased proteasomal activities were revealed in the SN of PD patients [40] and in cortical areas of DLB patients [41], and these are thus the brain regions mostly affected by α-syn pathology. In addition, α-syn might influence the expression levels of other proteasomal components. Accordingly, loss of 26/20S proteasome α-subunits and loss of the 19S RP were detected in the SN of PD patients, while the expression of 19S RP α-syn pathology unaffected brain regions of the same patients was even increased [40]. In DLB, a reduction in the Rtp6/S8 subunit of 19S RP in different cortical areas was linked to cognitive impairment [41].

Similarly, the dysregulation of chaperone-mediated autophagy (CMA) as well as macroautophagy is thought to underlie the pathogenesis of α-synucleinopathies. Studies on human postmortem brain tissue have shown the colocalization of the microtubule-associated protein 1A/1B-light chain 3 (LC3), a marker for autophagic vacuoles, with accumulated α-syn and LBs in the SN of PD patients [42] and in vulnerable brain regions in DLB patients [43]. More interestingly, the levels of the lysosomal-associated membrane protein type 2A (LAMP2A), lysosomal luminal HSC70 [42] as well as of LAMP2 [43] are decreased in these brain areas, suggesting alterations of the autophagic flux. Moreover, Beclin-1, a central regulator of autophagy, was found to be caspase-cleaved and consequently inoperable in LBs and apoptotic cells within the SN of postmortem PD and DLB brains [44].

Even though α-syn is considered a cytosolic protein, neurons can also secrete different α-syn species into the brain’s extracellular environment [45]. Secreted α-syn can be detected in human plasma and in the cerebrospinal fluid (CSF) [46]. For the CNS, it is assumed that not only extracellular proteolytic enzymes and astrocytes, but also mainly microglia, phagocytose extraneuronal α-syn [47] to prevent further spreading to neighboring neurons. More recently, Choi et al. have reported that neuron-released wild-type α-syn is cleared by microglia via selective autophagy, which is mediated by microglial TLR4 (toll-like receptor 4)-NF-κB (nuclear factor kappa B) signaling and the subsequent transcriptional upregulation of the autophagy receptor p62 [48].

Indeed, aggregated α-syn also has a direct effect on microglial activity, which is linked to the increased release of proinflammatory mediators [49] as well as an altered clearance process [47].

## 5. Microglia-Mediated Neuroinflammation in PD and DLB

Microglia, the resident innate immune cells of the CNS, make up approximately 10–15% of all cells in the brain, where they participate in a variety of processes that are necessary for maintaining tissue homeostasis [50]. Microglia are important modulators of neuronal differentiation and survival by secreting growth factors and trophic factors [51]. In addition, they regulate neuronal activity through synaptic remodeling [52] and are involved in the induction of apoptosis and removal of defective neurons and glia cells [53]. In the healthy brain, microglia are described to appear in a resting state, in which they are stationary and feature a highly ramified phenotype for immune surveillance. Contrary to their name, resting microglia are, in fact, quite active and constantly survey their environment with their ramified processes [54]. Upon the detection of stimuli generated by injury, degenerative events or microbial infection, microglia can thus respond quickly and transform into a reactive state. Reactive microglia change their transcriptional profile, start to produce inflammatory cytokines as well as chemokines and alter their receptor expression pattern to facilitate cytokine communication between cells [55]. Besides their morphological change to an amoeboid shape, activated microglia migrate towards the stimulus and become proliferative and highly phagocytic [56].

Elevated levels of microglia-secreted proinflammatory factors such as Interleukin-1β (IL-1β), Interleukin-6 (IL-6) and tumor necrosis factor α (TNFα), and the enhanced production of nitric oxide (NO) and ROS, can cause neuronal damage, especially when microglia become chronically activated [57]. This complex mechanism is summarized under the umbrella term neuroinflammation and is a common hallmark of various neurodegenerative diseases [58], suggesting that neuroinflammation might be the central modulator of disease progression.

Neuroinflammation is therefore an inflammatory condition of the nervous system, mediated by the brain’s innate and adaptive immune system, and can be defined on a molecular and cellular level. Indeed, elevated levels of IL-1β, IL-6, TNFα and eicosanoids are confirmed in nigrostriatal brain regions and the CSF of PD patients [59,60,61]. In addition, immunohistochemical analyses have revealed the increased expression of inducible nitric oxide synthase (iNOS) in glial cells within the SN of PD patients, which hints at the neurotoxic overproduction of NO and other free radicals, especially in the microglia of the parkinsonian brain [62,63]. Moreover, the possible involvement of adaptive immune responses in PD and DLB was demonstrated by T lymphocyte brain infiltration [64,65,66], which might be related to the antigen-presenting properties of microglia [64].

The extent to which microglial activation and neuroinflammation contribute to the development or progression of α-synucleinopathies remains a matter of debate. Widespread microgliosis was detected in the brains of PD patients as well as in the brains of subjects with early stages of DLB by ^11^C-PK11195 positron emission tomography (PET) imaging of the microglial activation marker translocator protein (TSPO) [67,68]. Lavisse and colleagues confirmed increased microglial activation in the nigrostriatal pathway and the frontal cortex of PD patents via ^18^F-DPA714 TSPO PET imaging [69]. PET imaging using the TSPO ligand ^11^C-PBR28, however, did not show elevated microglial activation in PD as compared to control subjects [70].

At the protein level, in both PD and DLB, microglia have previously been shown to have elevated levels of the major histocompatibility complex class II (MHCII) in brain regions affected by α-syn pathology, which implies an interaction of microglia with the adaptive immune system [71,72]. In the same brain regions, increased immunoreactivity against CD68, a marker for phagocytic microglia, was found in PD and DLB cases compared to control groups [71,73]. Nonetheless, it must be noted that these findings do not allow us to draw any pathogenic links between activated microglia and neurodegeneration in the concerned brain areas [71,72,73]. Indeed, a more recent human postmortem study on DLB could not find differences in the cortical protein load for either microglial activation or inflammation markers between DLB cases and matched controls [74]. In this respect, it should be considered that it is still not clearly understood how and at what stage of PD/DLB progression microglial responses become detrimental. For instance, PET imaging of DLB patients revealed elevated microglial activation in early disease stages with mild cognitive impairments, but not in full-blown DLB [68]. Thus, it is possible that the microglial response is only prominent in the early disease stages of DLB and fades as cognition declines. Similarly, PET imaging results suggest that microglial activity is already increased early in the disease process of PD and then remains stable over the years [67,69].

Findings of current gene expression studies further point out the involvement of microglia in the pathophysiology of PD/DLB [75,76]. Compared to healthy control brains, single-cell RNA sequencing discovered an enhancement of the microglia fraction in the anterior cingulate cortex of DLB and PD [75] patients, as well as in the PD midbrain [76]. In addition, microglia of the idiopathic PD midbrain contain subpopulations that are enriched towards an activated state [76].

## 6. The Role of Leukotrienes in CNS Inflammation and Neurodegeneration

Leukotrienes (LTs) are a group of highly active lipid inflammatory mediators constituted of five different types, namely leukotriene A4 (LTA_4_), leukotriene B4 (LTB_4_) and the cysteinyl leukotrienes (CysLTs) leukotriene C4 (LTC_4_), leukotriene D4 (LTD_4_) and leukotriene E4 (LTE_4_). In the human body, LTs are primarily synthesized by leukocytes from arachidonic acid (AA) via the so-called 5-lipoxygenase (5-lox) pathway [77]. After encountering an inflammatory stimulus, the substrate for LT production, AA, is provided by phospholipase A2 (PLA2) through cleavage from membrane phospholipids [77]. The released AA is converted into 5-hydroxyperoxyeicosatetaenoic acid (5-HPETE) and further into LTA_4_ by 5-lox. For these steps, the rate-limiting enzyme 5-lox is anchored to the nuclear envelope and activated by 5-lipoxygenase activating protein (FLAP) [78]. LTA_4_ is then either hydrolyzed to LTB_4_ by LTA_4_ hydrolase or conjugated with glutathione by LTC_4_ synthase, giving rise to LTC_4_ [77]. The cysteinyl leukotriene LTC_4_ may further be transformed into LTD_4_ and LTE_4_ through hydrolysis [77] (Figure 3).

In general, LTs exert immune responses by binding to G-protein-coupled receptors presented at the surfaces of target cells (e.g., smooth muscle cells, endothelial cells, mature hematopoietic cells [79,80]). A distinction is drawn between two classes of LT receptors: while LTB_4_ binds to the leukotriene B4 receptors BLTR_1_ and BLTR_2_ [81], the CysLTs bind with different affinities to cysteinyl leukotriene receptor 1 (CysLT_1_R; LTD_4_ > LTC_4_ > LTE_4_) and cysteinyl leukotriene receptor 2 (CysLT_2_R; LTC_4_ = LTD_4_ >> LTE_4_) [82]. However, as cysteinyl leukotriene receptors (CysLTRs) share homologies to members of the purinergic (P2Y) receptor family of G-protein-coupled receptors [83], other receptors, such as G protein-coupled receptor 17 (GPR17) and the platelet P2Y_12_ receptor (P2Y12R), are also suspected to be responsive to CysLTs [84,85].

The role of LT-mediated inflammation and its effects on different cell types was extensively studied in the pathophysiology of asthma [86,87], but emerging evidence suggests that LTs, their receptors and LT signaling might also be involved in neurodegenerative diseases [19,88,89,90,91].

Indeed, the CysLTRs, GPR17 and P2Y12R are present on different cell types of the human and rodent brain: under physiological conditions, CysLT_1_R is expressed in microglia and, to a lesser extent, in neurons [88,92], while CysLT_2_R can be found primarily in astrocytes [93]. GPR17 is mainly detected in neurons [84] and P2Y12R in microglia [94,95]. The upregulation of CysLTRs and/or GPR17 on neurons and glial cells was observed after brain trauma in rats [84,93] and humans [92], as well as on microglia in the brains of old rats [88].

Previous studies have indicated that LT signaling plays a crucial role in neuroinflammation by regulating the activity of microglia and astrocytes. For instance, in vitro experiments have shown that LTD_4_-induced microglial activation is mediated via CysLT_1_R [96], CysLT_2_R [97] and GPR17 [98]. A more recent in vitro study indicates that also LTB_4_ can promote the activation of microglia via BLTR_1_ [99]. By contrast, CysLT_1_R seems to be the main mediator of astrocyte activation [100,101]. It is assumed that glial cells and neurons are capable of producing LTs and that the release of LTs by microglia and astrocytes drives neuroinflammation in an autocrine fashion [99,101,102,103].

## 7. The Leukotriene Signaling Pathway in CNS Disorders

Involvement of the LT signaling pathway in the CNS and its pharmacological inhibition in vivo have previously been described, especially in the context of AD (reviewed in [19]). An immunohistochemical study of the human postmortem brain demonstrated that the 5-lox enzyme is elevated surrounding neurofibrillary tangles and amyloid beta (Aβ) plaques in the hippocampus of AD patients, where it likely contributes to increased LT production and neurodegeneration [104]. The putative role of 5-lox in the development of AD was confirmed in the triple transgenic (3xTg) mouse model of AD, in which neuronal overexpression of 5-lox led to the exacerbation of tangle and plaque pathologies, as well as memory deficits [105]. The genetic ablation of 5-lox, on the other hand, significantly reduced amyloid deposition in the brain in a Tg2576 AD mouse model [106]. The accumulation of Aβ is thought to be mediated by the increased activation of γ-secretase, which may be modulated by enhanced levels of 5-lox and its metabolite, LTD_4_ [106,107]. This was demonstrated in an in vivo study in which the intracerebral infusion of LTD_4_ in mice upregulated not only the expression of CysLT_1_R but also the activity of γ-secretase and thus increased the Aβ burden, leading to memory impairments [107].

To date, the role of LT signaling and signaling-related molecules in α-synucleinopathies has not yet been widely explored and data on LT production in humans are missing. Nevertheless, several rodent studies provide evidence for the possible involvement of the LT signaling cascade in the pathogenesis of PD and DLB [89,108,109].

First, the analysis of 5-lox expression in an α-syn transgenic mouse model of DLB (D-Line) revealed moderately elevated 5-lox levels in the brain compared to wild-type littermates [89]. Similarly, higher 5-lox expression was observed in the hippocampal neurons of human DLB patients compared to age-matched controls [89]. Thus, the LT production machinery seems to be induced in α-synucleinopathies.

Besides the above, LTs appear to be involved in neurodegeneration. In the context of PD, the 5-lox enzyme was found to affect the vulnerability of neurons in the nigrostriatal pathway in response to the selective dopaminergic toxin 1-methyl-4-phenyl-1,2,3,6-tetrahydropyridine (MPTP) [109]. In wild-type C57BL6 mice, injections with MPTP usually cause a substantial loss of dopaminergic neurons, as demonstrated by reduced striatal values of dopamine (DA) and its rate-limiting synthesis enzyme tyrosine hydroxylase (TH) compared to saline-injected C57BL6 mice [108,109]. Mice deficient in 5-lox expression were found to have lower striatal levels of TH and DA than wild-type littermates but appeared to be protected against MPTP-induced neurotoxicity, as shown by unchanged striatal DA levels compared to saline-treated 5-lox-deficient mice, suggesting that LTs contribute to neuronal cell death in pathological situations [109].

In addition to the role of LTs in neuronal cell death in the context of α-synucleinopathies, there is accumulating evidence for a role of LTs in neuroinflammation and glial cell activation. MPTP injections caused enhanced expression levels of 5-lox and FLAP as well as increased neuroinflammation, represented by higher immunoreactivity for the astrocyte marker glial fibrillary acidic protein (GFAP) and the microglia marker ionized calcium binding adapter protein 1 (Iba1) in the striatum of wild-type mice [108,109]. Kang et al. further showed the colocalization of 5-lox with GFAP, indicating that nigrostriatal damage by MPTP might, indeed, be related to gliosis [108]. However, toxin-challenged 5-lox-deficient mice show no signs of inflammation, indicated by the absence of microglia or astrocyte activation [109]. Together, these findings suggest that 5-lox contributes to physiologic striatal DA homeostasis but, when overactivated, leads to neuroinflammation and contributes to neurodegeneration.

It is not at all understood how neuroinflammatory and neurodegenerative events are mediated farther downstream of 5-lox activation in glial cells. Amongst the 5-lox metabolites downstream of AA, LTB_4_ but not LTD_4_ or 5-HETE (5-hydroxyeicosatetraenoic acid) enhanced the dopaminergic neurotoxicity in primary rat midbrain neuron–glia co-cultures [108]. Interestingly, neuronal survival could be restored by MK-886, a potent suppressor of LT biosynthesis that inhibits FLAP [108]. Other in vitro studies examining the role of 5-lox signaling in activated microglia provide evidence that CysLTs and their receptors, CysLT_1_R and CysLT_2_R, are involved in mediating neurotoxicity [110,111,112]. In these experiments, murine BV2 microglia cells were activated with lipopolysaccharide (LPS) or the pesticide rotenone. The cells’ medium was reused for the cultivation of rat neuronal cells (PC12) to evaluate microglia-dependent neurotoxicity [111,112]. Conditioned media from LPS-treated and rotenone-treated BV2 cells significantly decreased the viability of PC12 cells, as measured by 3-(4,5-dimethylthiazol-2-yl)-2,5-diphenyltetrazolium bromide (MTT) assays. Immunohistochemical and enzyme-linked immunoassay (ELISA) analyses revealed that rotenone-stimulated BV2 microglia not only released proinflammatory cytokines into their culture medium but also featured upregulated expression of 5-lox and CysLT_1_R and increased production of endogenous CysLTs [110]. The pretreatment of BV2 microglia with both the 5-lox inhibitor zileuton and the CysLT_1_R antagonist Montelukast (MTK), however, counteracted the rotenone-induced microglial activation and cytokine release [110]. In a follow-up study, zileuton pretreatment could prevent the previously observed microglia-mediated rotenone toxicity in PC12 cells [111]. Chen et al. detected increased expression levels of CysLT_2_R in LPS-stimulated BV2 cells and were able to attenuate the cytokine overproduction as well as the toxic effects of their conditioned medium on PC12 cells by the CysLT_2_R-selective antagonist HAMI 3379 [112].

These in vitro findings give sufficient reason to hypothesize that the 5-lox pathway might be a key regulator of glial cell activation and is consequently implicated in mechanisms of neuronal loss in the context of neurodegenerative diseases. Although the contribution of LTs specifically in the pathogenesis of PD and DLB remains elusive, LT signaling and its inhibition are potential new treatment strategies.

## 8. The Contribution of Leukotrienes in Autophagic Activity Impairment and Mitochondrial Dysfunction Outside the CNS

Although the impact of LT signaling on autophagy or mitochondria is largely unresearched in neurodegenerative diseases, studies on non-neurological conditions suggest detrimental effects of LTs on the autophagic and mitochondrial system [113,114,115,116,117,118].

In a rat model of aluminum (Al)-induced chronic liver injury, the 5-lox pathway was found dysregulated, as represented by enhanced activity of 5-lox [113], as well as increased expression levels of CysLTs, CysLTRs and BLTR_1_ [114]. Moreover, Western blot analysis revealed the upregulation of p62 under hepatotoxic conditions and downregulation of the autophagic vacuole marker LC3BII, indicating that autophagy is inhibited in the Al-overloaded rat liver. The impairment of autophagy was confirmed via Western blot and transmission electron microscopy in Al-induced human hepatocyte line L02, and blocking of CysLT_1_R via MK-571 could restore autophagy and reduce the cell death rate [114]. The potential role of CysLT_1_R in modulating autophagic processes was additionally shown in retinal pigment epithelial cells, in which CysLT_1_R inhibition by Zafirlukast treatment in the presence of lysosomal inhibitors led to higher LC3BII protein levels, indicative of increased autophagic flux, compared to the untreated control group [115].

Notably, LT signaling seems to affect mitochondrial function at the CysLT_1_R as well as at the BLTR_1_ level [116,117,118]. On the one hand, the intraperitoneal administration of the BLTR_1_ inhibitor U75302 reversed mitochondrial complex I and II downregulation and attenuated mitochondrial impairment in the myocardium of an LPS-induced acute cardiac injury mouse model [116]. On the other hand, the idea of a possible therapeutic effect of LT inhibition on mitochondria is supported by in vitro studies on human bronchial epithelial cells showing that the CysLT_1_R antagonists Zafirlukast and MTK stimulate mitochondrial biogenesis via PGC-1α [117,118].

With regard to α-syn pathology, however, the role of LTs in the autophagic and mitochondrial machinery requires more detailed investigation.

## 9. Targeting Leukotriene Signaling in Preclinical Studies

Reports of beneficial effects through the pharmacological modulation of LT signaling in AD animal models further support the potential of the LT system as a druggable target in CNS diseases (reviewed in [19]). For instance, the administration of the 5-lox inhibitor zileuton for 10 months through drinking water not only reduced Aβ and phosphorylated tau pathology in 3xTg AD mice but also ameliorated their behavioral impairments in the Y-maze, Morris water maze (MWM) and fear conditioning tests [119]. Improved cognitive functions were also observed in rodent AD models by antagonizing CysLT_1_R via Pranlukast, Zafirlukast or MTK [91,120,121]. The learning and memory decline observed after bilateral hippocampal injection of Aβ_1-42_ in mice could be reduced by a 4-week daily oral treatment with Pranlukast [120]. Likewise, in an intracerebroventricular Aβ_1-42_ oligomer-induced experimental AD rat model, the intraperitoneal (i.p.) administration of Zafirlukast for 21 days improved cognitive deficits, as assessed by the MWM and novel object recognition test [121]. In addition, the attenuation of behavioral deficits in the MWM test was observed especially in female 5xFAD transgenic mice after daily oral treatment with MTK for 89 days [91].

The aforementioned studies on CysLT_1_R antagonism in AD explained the observed improved cognitive performance not only by the attenuation of apoptotic and inflammatory responses [120], including the modulation of microglia and CD8+ T-cells [91], but also by restoring mitochondrial respiratory enzyme complex activities [121]. Considering that impaired mitochondrial dynamics are largely involved in neurodegeneration, the mito-protective effects of CysLT_1_R inhibition could be vitally important for the development of new strategies against CNS diseases. Interestingly, Kumar et al. showed that a 2-week daily i.p. MTK treatment in a kainic acid-induced cognitive dysfunction rat model improved memory performance, which might be traced to the restoration of mitochondrial functions [122]. An attenuation of mitochondrial dysfunction was, inter alia, also observed in a study on quinolinic acid/malonic acid-induced striatal neurotoxicity in rats after daily oral treatment with MTK for 21 or 14 days, respectively [123].

Focusing on α-synucleinopathies, the impact of LT signaling blockage has, in the last few years, also been assessed in vivo using different animal models of PD and DLB. As an example, the inhibition of LT biosynthesis at the 5-lox level via i.p. injection of the selective FLAP inhibitor MK-886 for 7 days not only attenuated LTB_4_ production, but also decreased dopaminergic neuronal cell death in the striatum and SN of an MPTP-induced mouse model for PD [108]. Interestingly, a study using human granulocytes suggests that phenolic compounds and flavonoids, which are present in a variety of spices, exert inhibitory effects on cellular 5-lox activity [124]. This seems also to apply to cells of the CNS, since the i.p. administration of an extract from the pepper *Capsicum annuum* L. was shown to alleviate the increased 5-lox activity in the brain of a rotenone-induced PD mouse model [125]. Furthermore, the extract prevented neuronal injury caused by rotenone in the SN, cerebral cortex and hippocampus in a dose-dependent manner, most probably by interfering with LT production via the 5-lox pathway [125].

To analyze the effects of intervening LT signaling at the receptor level in PD and DLB, recent animal model studies almost exclusively focused on the drug MTK. For this reason, the next section will cover studies with MTK and their findings in more detail.

## 10. Montelukast as a Treatment Option for PD and DLB

MTK is an approved selective CysLT_1_R antagonist used for the treatment of asthma as well as allergic rhinitis [77]. Its primary mode of action is the inhibition of inflammatory signaling by CysLTs through CysLT_1_R. However, MTK also antagonizes GPR17 [84] and was found to possess direct inhibitory effects on 5-lox in vitro, presumably through binding of an allosteric site, resulting in reduced LTB_4_ production [126,127,128].

It does not seem unreasonable to consider MTK for repurposing as a treatment approach for PD and DLB. In this context, analyses of a Norwegian Prescription Database (NorPD) show that patients that were prescribed the asthma medicine MTK lived longer and had a lower chance of developing dementia than users of popular inhalation medicine (mainly inhaled corticosteroids). Moreover, these patients achieved better scores in brain function tests compared to users of other anti-inflammatory drugs. This suggests that MTK may alleviate the age-related cognitive decline [129,130]. As a matter of fact, MTK is currently being tested in clinical trials for use in AD (NCT03991988 and NCT03402503) as well as in PD (EudraCT: 2020-00148-76). The investigation of the NorPD using a regression model revealed a reduced risk of developing PD associated with the asthma drug salbutamol, a β2-adrenoreceptor agonist, which is discussed to regulate the transcription of the SNCA gene [131]. A study investigating the effects of salbutamol injection in guinea pigs exposed to dry-gas hyperpnea challenge found that their biliary release of LTD_4_ was inhibited [132]. Although this is a very vague conclusion, a link between the reduced PD risk in salbutamol users and a possible modulation of LT synthesis by salbutamol could be considered.

In a 6-hydroxydopamine (6-OHDA)-induced PD mouse model, the i.p. administration of MTK for 7 consecutive days was reported to diminish microglial cytokine release as well as 6-OHDA-mediated neurotoxicity in the striatum and SN [133]. Despite the alleviated inflammatory response and protection of DA neurons by MTK, the treatment did not ameliorate the anomalous locomotor behavior of these mice, as assessed by the rotarod test [133]. Neuroprotective effects were also observed in the cerebral cortex and SN in a rotenone-induced rat model of PD after 2 weeks of systemic MTK administration [134]. Furthermore, MTK was able to attenuate the rotenone-evoked oxidative stress in the rat brain, as indicated by a decrease in nitric oxide and malondialdehyde (MDA) concentrations as well as an increase in the level of the antioxidant glutathione (GSH) [134]. Another study using a rotenone rat model of PD confirmed the antioxidative effects of MTK after 2 weeks of treatment, as shown by reduced MDA levels and enhanced GSH levels in brain tissue homogenates [135]. This study also evaluated the impact of MTK on rotenone-induced motor impairment and reported improved motor activity in the open field test as well as improved performance in the catalepsy and rotarod tests in MTK-treated rats compared to vehicle-treated animals [135].

Inhibition of LT signaling remains widely unexplored in the field of DLB; however, recently, the effects of MTK have been assessed in an α-syn transgenic mouse model (D-Line) [89]. This genetic model overexpresses human α-syn under the regulation of the platelet-derived growth factor β (PDGF-β) promoter and features neuronal α-syn immunoreactive inclusions as well as cognitive deficits that precede motor impairments [136,137]. The D-Line mouse model can thus be considered a model for DLB rather than a model of PD [113], [114]. Marschallinger and Altendorfer et al. not only demonstrated a reduction in the α-syn load, but also increased expression of beclin-1 in the brains of D-Line animals after daily administration of MTK for 6 weeks [89]. This suggests that MTK might support protein clearance via stimulation of macroautophagy. In addition, MTK-treated D-Line mice showed improved memory compared to the vehicle-treated animals, which was assessed by the MWM spatial learning and memory test [89]. To date, there are no other reports on in vivo studies that test the effects of MTK in transgenic mice for DLB.

## 11. Concluding Remarks

Over the last few years, a considerable amount of evidence was collected underpinning the involvement of dysregulated LT signaling in various pathological processes of neurodegenerative diseases, including PD and DLB. The pharmaceutical intervention of the LT synthesis pathway was shown to have beneficial effects on different aspects of the pathogenesis, in vitro as well as in vivo. Together, all these data indicate that the repurposing specifically of the anti-asthmatic drug MTK would be a reasonable approach for the treatment of neurodegenerative diseases.

The focus of the in vivo studies performed so far to modulate the LT signaling system in α-synucleinopathies was previously laid on neuroinflammation and neurodegeneration, mainly using toxin-based models for PD. These animal models were designed to mimic nigrostriatal dopaminergic damage through the application of environmental toxins but are often criticized due to the fast onset of symptoms, which does not correspond to the gradual progression observed in PD patients [138]. Furthermore, it has to be considered that neurotoxic models such as the MPTP-induced model often do not encompass motor symptoms as observed in PD patients [139].

Indeed, previous in vivo studies have only sparsely addressed the impact of LT signaling blockage on behavior and/or motor functions in PD or DLB animal models and, even more importantly, almost completely neglected α-syn pathology, which is the defining hallmark of both diseases.

Only very recently, the effects of LT signaling inhibition on α-syn inclusions and behavior have been assessed in a transgenic α-syn mouse model [89]. Common genetic models of PD express wild-type or mutated forms of α-syn under a variety of different promoters and, in contrast to toxin models, provide a more reliable insight into the molecular pathogenesis underlying PD [140]. Although genetic models often fail to reproduce DA cell loss, they display intracellular α-syn inclusions, gliosis and functional abnormalities in the nigrostriatal system and exhibit motor deficits and sometimes even non-motor symptoms (reviewed in [140]). The use of transgenic α-syn animal models might hence cover a broader range of mechanisms potentially involved in the development of PD and DLB and would be reasonable for future drug candidate screenings. Apart from the D-Line mouse model, another well-established mouse model of the α-syn pathology is, for example, the Line 61. The Line 61 is a transgenic mouse model overexpressing α-syn throughout the brain under the regulatory control of the Thy-1 promoter [141]. Line 61 mice feature the early onset of motor impairments, which can be detected already at an age of 1 month, and additionally show signs of neuroinflammation, as indicated by subcortical astrogliosis starting from an age of 6 months [141]. Thus, it might be of interest to investigate the role of LT signaling and its inhibition in the Line 61 model.

Clearly, to determine the pathophysiological role of LT signaling in α-synucleinopathies, further research is required. Nevertheless, existing research results provide an extensive amount of evidence suggesting that targeting the LT signaling pathway might be a rational approach for the management of PD and DLB as well as other related neurodegenerative diseases. Obviously, a number of important issues in the design of clinical trials testing MTK in PD and/or DLB must be addressed. For example, what should be the clinical outcome parameters in such trials? Based on the mode of action and on the available preclinical data (Figure 4), it is conceivable that treatment with MTK might affect not only motor and cognitive outcomes, but also autonomic functions, sleep disturbances as well as neuropsychiatric features such as depression and anxiety. Therefore, a broad spectrum of outcome parameters should be considered. A further point is that, most likely, MTK needs to be tested as an add-on therapeutic agent in addition to the standard medication—for example, levodopa in the case of PD. Since the mode of action of MTK is different from the standard medications, an additive effect could be expected.

To summarize, it can be assumed that leukotriene modulators would cover additional mechanisms that, alongside existing symptomatic medications, would provide a broader spectrum of beneficial effects and eventually improved efficacy in PD/DLB patients.

## Figures and Tables

**Figure 1 biomolecules-12-00346-f001:**
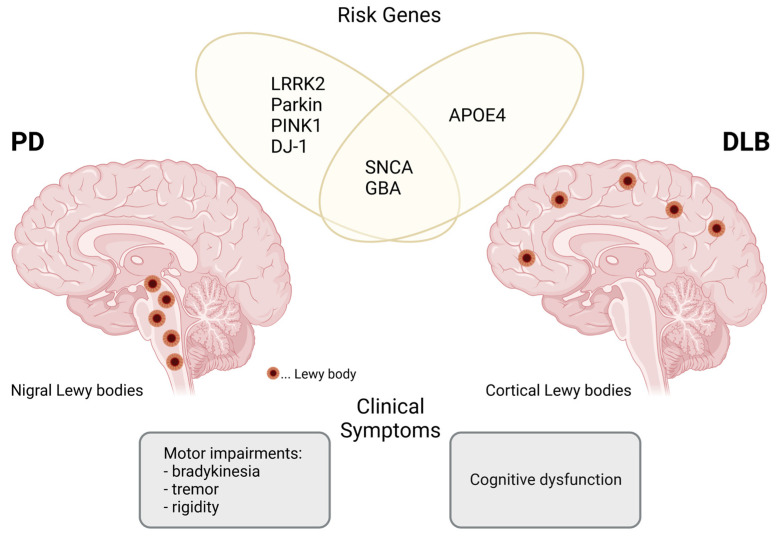
Clinical differentiation between Parkinson’s disease (PD) and dementia with Lewy bodies (DLB) and common genetic determinants. In the PD brain, aggregated α-synuclein and Lewy bodies (depicted as red circles) are mainly located in the substantia nigra and the brainstem, which is accompanied by dopaminergic neuronal death, causing typical parkinsonian motor symptoms. In DLB, α-synuclein accumulations and Lewy bodies are predominantly found in cortical brain areas involved in cognitive impairments. PD and DLB both feature individual predisposition genes (PD: LRRK2, Parkin, PINK1 and DJ-1; DLB: APOE4), but also share risk genes that influence the aggregation of α-synuclein (e.g., SNCA and GBA). Created with BioRender.com (last accessed on 1 January 2022).

**Figure 2 biomolecules-12-00346-f002:**
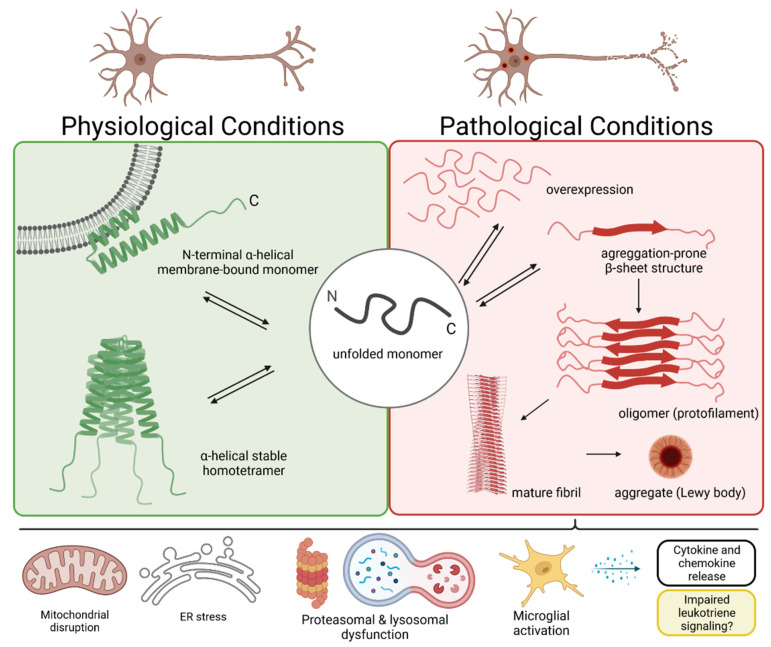
Structural features of functional and dysfunctional α-synuclein. Physiologically, α-synuclein likely exists in a dynamic equilibrium between an unfolded monomer and an α-helical membrane-bound or homotetrameric structure. Both the overexpression as well as the conformational change into a β-sheet structure increase the susceptibility of α-synuclein to accumulate. The aggregation process involves the formation of oligomeric protofilaments, which mature to fibrils and associate with other proteins to build up Lewy bodies. Under pathological conditions, aggregates of α-synuclein and Lewy bodies interfere with a variety of cellular processes, including mitochondrial and endoplasmic reticulum functions, protein degradation systems as well as with the activation state of microglia. Together, the disruption of cell homeostasis and the proinflammatory response of microglia (via releasing cytokines, chemokines and/or leukotrienes) eventually lead to neuronal cell damage and death. Created with BioRender.com (last accessed on 4 November 2021).

**Figure 3 biomolecules-12-00346-f003:**
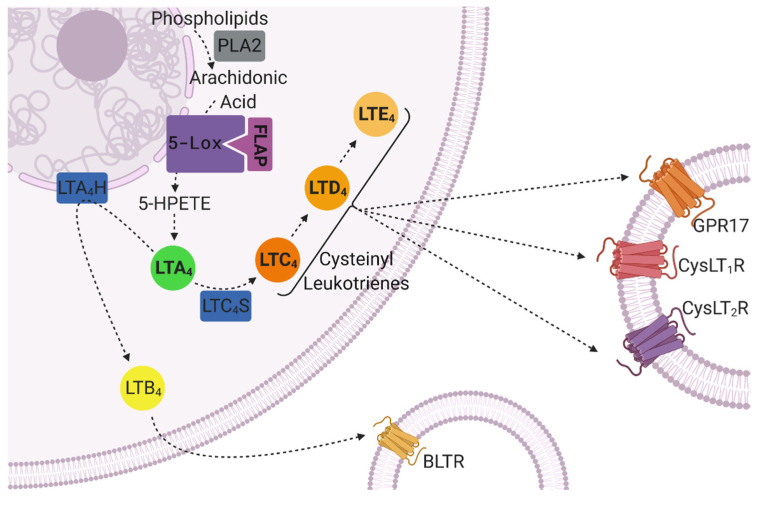
The leukotriene biosynthesis and signaling pathway. Leukotrienes derive from arachidonic acid, which is cleaved from membrane phospholipids by phospholipase A2 (PLA2) and further processed by the key enzyme 5-lipoxygenase (5-Lox) and its activator, 5-lipoxygenase activating protein (FLAP). The intermediate product 5-hydroperoxyeicosatetraenoic acid (5-HPETE) gives rise to leukotriene A_4_ (LTA_4_), which is further converted to leukotriene B_4_ (LTB_4_) by leukotriene A_4_ hydrolase (LTA_4_H) or to the cysteinyl leukotrienes leukotriene C_4_, D_4_ and E_4_ (LTC_4_, LTD_4_ and LTE_4_) by leukotriene C_4_ synthase (LTC_4_S). Leukotrienes evoke inflammatory responses by binding with different affinities to specific receptors on target cells. These include the leukotriene B_4_ receptor (BLTR), the G-protein-coupled receptor 17 (GPR17), the cysteinyl leukotriene receptor 1 (CysLT_1_R) and cysteinyl leukotriene receptor 2 (CysLT_2_R). Created with BioRender.com (last accessed on 4 November 2021).

**Figure 4 biomolecules-12-00346-f004:**
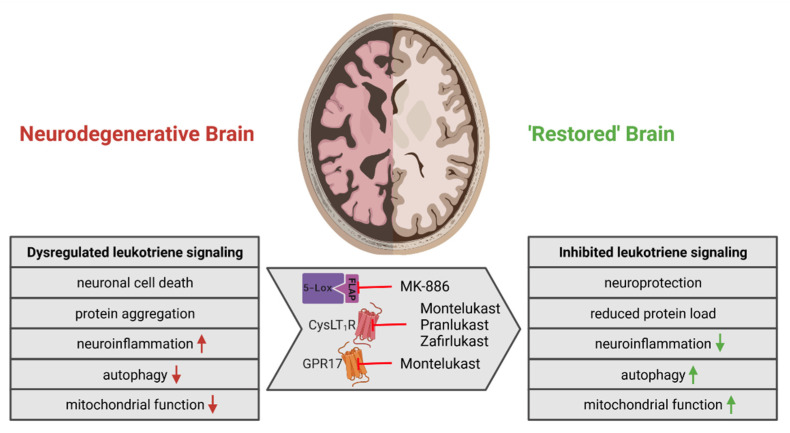
Overview of the effects of leukotrienes and leukotriene inhibition on the neurodegenerative brain. The interference of leukotrienes with various aspects and processes of brain homeostasis contributes to the development of neurodegenerative diseases. Several in vitro and in vivo studies have shown that the inhibition of leukotriene signaling at the level of 5-lipoxygenase by MK-886 or at receptor level using the cysteinyl-leukotriene receptor antagonists Montelukast, Pranlukast or Zafirlukast can ameliorate these impairments and lead to restored brain function. Upward arrows indicate an increase and downward arrows indicate a decrease. Created with BioRender.com (last accessed on 4 November 2021).

## Data Availability

Data are stored at the local PMU server and available upon request.

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
