# Peer review of "Leukotriene Signaling as a Target in α-Synucleinopathies"

_biomolecules, 2022, doi:10.3390/biom12030346_

Round 1

Reviewer 1 Report

The review ”Leukotriene Signaling as a Target in α-Synucleinopathies” by Strempfl and colleagues is timely, informative and nicely written and illustrated.

Minor points

  1. Lines 72-76: Albeit not the focus of this review, the authors should expand on approved symptomatic therapies for PD and, at least, mention MAO inh, COMT inh and DBS against along with choline esterase inh against DLB. In this context, the word “massive” on line 73 seems unfitting. Consider exchanging or rephrase the sentence.
  2. Lines 240-242: Preliminary studies with more recent TSPO ligands, such as [11C]PBR28, has not shown any difference between controls and PD (Varnäs K, Cselényi Z, Jucaite A, Halldin C, Svenningsson P, Farde L, Varrone A. PET imaging of [11C]PBR28 in Parkinson's disease patients does not indicate increased binding to TSPO despite reduced dopamine transporter binding. Eur J Nucl Med Mol Imaging. 2019 Feb;46(2):367-375. doi: 10.1007/s00259-018-4161-6). In contrast to the cited studies here using [11C]PK11195. This discrepancy should be mentioned.
  3. Line 246: Paragraph is a bit unclear,” In the same brain regions demonstrate PD as well as DLB cases elevated immunoreactivity against CD68, a marker for phagocytic microglia, when compared to control groups.” Consider changing the placement of the word “demonstrate” or rephrase completely.
  4. Line 428: Paragraph is a bit unclear, “The aforementioned studies on CysLT1R antagonism in AD, explained the observed improved cognitive performances not only by attenuation of apoptotic and inflammatory and responses including the modulation of microglia and CD8+ T-cells, but also by restoring mitochondrial respiratory enzyme complex activities.” Consider omitting one “and” or rephrase completely.
  5. In the section on Montelukast as a Treatment Option for PD and DLB, consider adding a comment on ongoing clinical studies. NCT03991988 and NCT03402503 for Alzheimer’s disease and EudraCT: 2020-000148-76 for PD.

Reviewer 2 Report

Review:

In the manuscript ‘Leukotriene signaling as a Target in α-synucleinopathies’ the authors provide an overview of PD pathology including the role of microglia and inflammation and introduce leukotrienes as interesting targets for further drug research. To this end, this manuscript features a proposal to repurpose medication already approved for asthma, targeting leukotrienes, to be used for PD patients.

The manuscript, especially the first 4 parts are very well written and were enjoyable to read, giving succinct but nevertheless clear overview of PD pathology. The major points are mostly related to the later part of the manuscript:

Major points:

1) This review relies heavily on the link of leukotrienes and (neuro)inflammation. However, the presence of neuroinflammation, and more specifically microglial activation in  DLB/PD brains is subjective to discussion. It remains questionable if global microglial activation and neuroinflammation in the DLB/PD brain are a primary or secondary response to α-synucleinopathy and at which stage of the disease inflammation may be relevant.  This is highlighted in some of the papers used for this review i.e. Reference [63] does not find that activated microglia (as indicated by increased expression of MHC-II) in the PD brain are related to neuronal degeneration or clinical symptoms (more specifically motor symptoms). In addition, reference [65] does describe increased CD68 in the DLB brain, but does not find a relation with neurodegeneration. In their discussion, the authors express caution in deducing from their data that there is increased microglial activation in the DLB brain compared to (aged) controls.

As some sources used in this review used as evidence of microglial activation in the DLB or PD brain are relatively old (i.e. 1994 [56, 57], 2000[64], 2016 [65]), it may be worth adding additional more recent studies. For example: a more recent study post-mortem study assessing microglial activation in DLB brains compared to control found no difference in microglial activation (Neuroinflammation in dementia with Lewy bodies: a human post-mortem study | Translational Psychiatry (nature.com))

However, this is not to say that there is no microglial response in the PD/DLB brain. More recent gene expression studies do highlight changed microglial gene expression in the PD(D) and/or DLB brain (i.e. Cross-platform transcriptional profiling identifies common and distinct molecular pathologies in Lewy body diseases | SpringerLink). Moreover PET studies do find a microglial response in DLB brains (as mentioned also by the authors). In addition there is ample evidence that microglia may play a role in spreading of α-syn throughout the brain.

Including more recent data on the role of microglia/neuroinflammation in PD or DLB brains and adding more discussion on the various roles of i.e. microglia in PD/DLB at different disease stages (i.e. disease onset, or late in the disease) may strengthen the review.

2) Though this review aims to link the role of leukotrienes to α-synucleinopathies (as is mentioned in the title of the report) proposing that repurposing of drugs targeting leukotrienes could be of interest for PD/DLB patients, there seems to be more discussion on the role of leukotrienes in neuroinflammation and neurodegeneration than the primary pathological and clinical symptoms of PD/DLB specifically. Papers used to highlight the role of leukotrienes in PD mouse models are primarily based on data from toxin-induced models which likely also induce heavy neuroinflammation (as also rightly mentioned by the authors).  Though there may be not sufficient data yet to discuss the effect of leukotrienes on PD/DLB specific pathology, the authors should nevertheless make it more clear why targeting leukotrienes may still be beneficial for PD/DLB patients and which outcome parameters should be taken into account when assessing the potential beneficial effects of leukotriene modulators in i.e. clinical trials. Which specific clinical symptoms do the authors hypothesize will be targeted? How do the authors think leukotriene targeting medication may perform relative to standard medication currently given? (i.e. L-DOPA?).

Minor points:

  • Page 6, line 232-233: Here it is stated that ‘Neuroinflammation is therefore […] mediated by the brains innate immune system […]’. However, the term neuroinflammation is not limited to inflammation caused by the innate immune system and can also be used in reference to inflammation related to the adaptive immune system. Related to this, there is evidence for activation of the adaptive immune system in PD and potential infiltration of lymphocytes in the PD or DLB brain.
  • Page 6, line 242-243: Here it is stated that there is evidence for ‘an altered gene expression profile’ but the studies subsequently described do not feature gene expression data but instead describe protein data (i.e. immunohistochemistry).
  • Page 8, line 239: To my knowledge, in the human brain, P2Y12 is expressed by microglia only, not by astrocytes. The reference used [84] does asses P2Y12 immunoreactivity in the brain but reports no co-localization of P2Y12 and GFAP.
  • Page 10, lines 403-406. In this section ‘on the other hand’ is used twice in a row.
  • Page 11, lines 467-473: Evidence used here to support a potential beneficial role of salbutamol in PD related to its potential ability to modulate LTD4 is rather weak and should be worded with a bit more caution than it is now. 

Reviewer 3 Report

Review of a manuscript “Leukotriene Signaling as a Target in α-Synucleinopathies” by Katharina Strempfl and coauthors submitted to “Biomolecules”.

Synucleinopathies are a group of severe neurodegenerative disease for which there is no efficient treatment yet and no reliable biomarkers for early identification. So basic research aimed at better understanding of molecular mechanisms underlying these disorders are important and timely. Leukotrienes play an important, although not completely understood role in the pathogenesis of synucleinopathies. The authors of this review manuscript put forward a hypothesis that dysregulation of leukotriene signaling is involved in pathological processes underlying some of synucleinopathies, especially Parkinson’s disease and dementia with Lewy bodies. This is an important biomedical field and the review will be interesting for the readers of “Biomolecules”.

The following corrections should be made.

Introduction

Line 51.  (Figure 1Error! Reference source not found.)

It is not clear what is this?

Line 52. “Not only SNCA, which is the gene encoding for α-syn” After this sentence the authors could consider adding a citation of a review on alpha-synuclein: “Synucleins and Gene Expression: Ramblers in a Crowd or Cops Regulating Traffic? Front Mol Neurosci. 2017; 10:224. doi: 10.3389/fnmol.2017.00224.”

Line 57. “Mutations associated with PD commonly occur in genes encoding…”. This should be corrected as follows :” Mutations associated with PD in addition to α-syn gene commonly occur in genes encoding…”

Line 66 “In the PD brain, aggregated α-synuclein and Lewy bodies…” Clarification needed to better understand the sense of the picture: “In the PD brain, aggregated α-synuclein and Lewy bodies (red circles)…”

Line 69 “Lewy bodies are predominantly found in cortical brain areas manifesting in cognitive impairments” should be corrected as follows “Lewy bodies are predominantly found in cortical brain areas involved in cognitive impairments”.

Line 70 “…but also shared risk genes that influence the aggregation of α-synuclein are known.” This iss a clumsy sentence the sense of which is unclear. What the authors want to say? May be  “…but also shared unknown risk genes that influence the aggregation of α-synuclein”

Lines 106-107 “The conformational change of a monomer into a β-sheet structure causes the transition of the physiological soluble α-syn into toxic insoluble deposits (Figure 2).” The authors could consider adding here a citation to “Analysis of Protein Conformational Strains-A Key for New Diagnostic Methods of Human Diseases. Int J Mol Sci. 2020; 21(8):2801. doi: 10.3390/ijms21082801.

Lines 321-322:”To date, the role of LT signaling in α-synucleinopathies has not yet been widely explored and data on LTs production and LT signaling related molecules in human patients are widely missing”. This is an awkward sentence. Can be corrected as follows: ”To date, the role of LT signaling and signaling related molecules in α-synucleinopathies has not yet been widely explored and data on LTs production in humans are missing.”

Concluding remarks

In concluding remarks the authors should give a short summary pointing to the most important findings. However a part of the concluding remarks, for example in lines 513-525 is just a continuation of the review data which can be deleted from conclusion or replaced to the main body of the review.
